# CANTO-RT: One of the Largest Prospective Multicenter Cohort of Early Breast Cancer Patients Treated with Radiotherapy including Full DICOM RT Data

**DOI:** 10.3390/cancers15030751

**Published:** 2023-01-25

**Authors:** Thomas Sarrade, Rodrigue Allodji, Youssef Ghannam, Guillaume Auzac, Sibille Everhard, Youlia Kirova, Karine Peignaux, Philippe Guilbert, David Pasquier, Séverine Racadot, Céline Bourgier, Sandrine Ducornet, Fabrice André, Florent De Vathaire, Sofia Rivera

**Affiliations:** 1Radiotherapy Department, Gustave Roussy, 94805 Villejuif, France; 2Department of Radiation Oncology, Tenon Hospital, Paris Sorbonne University, 75020 Paris, France; 3Unit INSERM UMR U1018, Gustave Roussy, 94805 Villejuif, France; 4UNICANCER, 75013 Paris, France; 5Institut Curie, 75248 Paris, France; 6Centre Georges-François Leclerc, 21079 Dijon, France; 7Institut Jean Godinot, 51100 Reims, France; 8Centre Oscar Lambret, 59020 Lille, France; 9Centre Léon Bérard, 69008 Lyon, France; 10ICM, 34298 Montpellier, France; 11Aquilab, 59120 Loos, France; 12Gustave Roussy, 94805 Villejuif, France; 13Inserm UMR 1030, Molecular Radiotherapy and Therapeutic Innovation, Paris-Saclay University, 94805 Villejuif, France

**Keywords:** early breast cancer, radiotherapy, toxicities’ predictive factors

## Abstract

**Simple Summary:**

Radiation therapy (RT) is one of the corner stones of the local treatment of breast cancer (BC). Toxicity factors related to RT and their consequences are poorly known because of limited DICOM data and limited analyses on contouring, dose distribution and the RT technique. This manuscript describes the methodology used and provides the first characterization of the study population and RT data in CANTO-RT (CANcer TOxicities RadioTherapy). To our knowledge, our study is the largest available multicenter prospective multicenter cohort of early breast cancer with full DICOM RT data (files (CT, RT Structure, RT Dose, RT Plan)). This study answers to a concern about toxicity factors related to radiotherapy and their consequences and aims to identify predictors of development and the persistence of long-term toxicities in breast cancer patients. Further long-term projects (heart, lung, skin, fatigue) and follow up is ongoing.

**Abstract:**

This article describes the methodology used and provides a characterization of the study population in CANTO-RT (CANcer TOxicities RadioTherapy). CANTO (NCT01993498) is a prospective clinical cohort study including patients with stage I-III BC from 26 French cancer centers. Patients matching all CANTO inclusion and exclusion criteria who received RT in one of the 10 top recruiting CANTO centers were selected. Individual full DICOM RT files were collected, pseudo-anonymized, structured and analyzed on the CANTO-RT/UNITRAD web platform. CANTO-RT included 3875 BC patients with a median follow-up of 64 months. Among the 3797 patients with unilateral RT, 3065 (80.4%) had breast-conserving surgery, and 2712 (71.5%) had sentinel node surgery. Tumor bed boost was delivered in 2658 patients (68.5%) and lymph node RT in 1356 patients (35%), including internal mammary chain in 844 patients (21.8%). Most patients (3691 (95.3%)) were treated with 3D conformal RT. Target volumes, organs at risk contours and dose/volume histograms were extracted after quality-control procedures. CANTO-RT is one of the largest early BC prospective cohorts with full individual clinical, biological, imaging and DICOM RT data available. It is a valuable resource for the identification and validation of clinical and dosimetric predictive factors of RT and multimodal treatment-related toxicities.

## 1. Introduction

Breast cancer (BC) is the leading cancer in women throughout the world. with 2.3 million new cases diagnosed and 685,000 related deaths in 2020 [1]. Efforts over the last two decades to reduce breast cancer mortality focus on early detection and treatment [2]. About 80% of breast cancer patients can expect long-term disease-free survival. In industrialized countries, about 5 million women live with a history of breast cancer and are at risk of facing treatment for long-term toxicity. Post-cancer is therefore an important part of their lives [3]. It has become a priority to reduce treatment-related toxicities in the management of breast cancer patients. Radiation therapy (RT) is one of the corner stones of local treatment of BC. Various meta-analyses of long-term follow-up have demonstrated an overall survival benefit from radiotherapy (RT) [4,5]. However, toxicity factors related to RT and their consequences are poorly known because of limited DICOM (Digital Imaging and Communications in Medicine) data with limited analyses on contouring (target and organs at risk volumes), dose distribution, RT technique and quality involving precise calculation and delivery of the planned dose. This understanding is nevertheless essential to characterize radiation-induced toxicities, to better understand treatment related toxicities and to identify the predictive factors for the occurrence of these toxicities. CANcer TOxicities (CANTO) (NCT01993498, UNICANCER 0140/1103, 2011-A01095-36 (‘study of chronic toxicity of treatment of patients with localized breast cancer’) is a multicenter prospective cohort study with the primary objective of identifying factors predictive of chronic toxicity in patients treated for stage I–III breast cancer [6]. Within CANTO, detailed RT data were collected for a subset of patients representing CANTO-RT (CANcer TOxicities RadioTherapy), a large multicenter prospective cohort of early breast cancer (BC) patients treated with RT that aims to identify predictors of the development and persistence of long-term toxicities. In this paper, we describe the methodology used to collect RT data (full DICOM) and to ensure RT data quality control to provide a first characterization of the study population and RT data in CANTO-RT.

## 2. Materials and Methods

### 2.1. Study Design 

CANTO (NCT01993498) is a French prospective longitudinal multicenter cohort study designed to evaluate chronic toxicities in patients treated for non-metastatic BC diagnosed and enrolled between 2012 and 2018, in 26 French centers. The details on the CANTO study procedures have previously been published in accordance with the French national regulatory requirements, good clinical practice guidelines and European General Data Protection Regulation (GDPR) as previously described by Vaz et al. [6]. This study, sponsored by Unicancer, enrolled 12,012 patients. In the database lock of December 2020, data from 2012 to 2017 were obtained corresponding to 9599 patients.

### 2.2. Study Population

The subset of patients matching all CANTO inclusion and exclusion criteria, who received RT in one of the 10 top recruiting CANTO centers with a minimum follow up of 3 years, and who were still in follow up at the time of the database lock were selected for CANTO-RT (Figure 1 Flowchart). Patients included were followed for 10 years as part of the study, with a minimum of 36 months follow-up. CANTO-RT patients met the following inclusion criteria: female patients aged 18 years and over covered by the national social security system, with histologically proven non-metastatic invasive BC (cT0-3, cN0–3) without previous cancer treatment. Conventional or hypofractionated RT was prescribed according to local standard-of-care. Eligible patients had breast/chest wall +/− lymph node RT with curative intent.

### 2.3. Data Collection 

Patients and multimodal treatment characteristics as well as paraclinical parameters including blood chemistries, exams, or toxicity data, etc., were collected prospectively (Figure 2) and were the same as described in [6]. Patients were assessed at diagnosis (baseline), 3–6 (M0), 12 (M12), 36 (M36) and 60 (M60) months after completion of chemotherapy or RT, whichever came last. In this study, radiotherapy data were exported in standardized Digital Imaging and Communications in Medicine (DICOM) format by each investigating hospital to the UNITRAD online platform hosted by AQUILAB Onco Place™, a company with health-data-hosting authorization. All data were automatically pseudo-anonymized and converted to homogeneous naming. We prospectively assessed data at diagnosis (baseline), 3–6 (M0), 12 (M12), 36 (M36) and 60 (M60) months after completion of chemotherapy or RT, whichever came last. Organizational structure was previously described [6] and a summary of the data collection is presented Figure 2.

In CANTO RT, individual full DICOM RT data (CT, RT Structure, RT Dose, RT Plan) were collected, pseudo-anonymized, structured and analyzed on the CANTO-RT/UNITRAD web platform using AQUILAB Onco Place™ and Analytics Dose module (Figure 3). In the Analytics Dose module, RT data were extracted, filtered and grouped according to sets of constraints by volumes (mean dose, median dose, DX%: dose covering X % of the volume expressed in Gy, VX Gy: volume receiving at least X Gy expressed in %, near-min dose, near-max dose).

We collected the platform RT data, the treated side (right, left, bilateral), whether or not there was the presence of a tumor bed boost, lymph node levels treated (none, level 1 to 4, interpectoral, Internal mammary chain), techniques (3D, IMRT: intensity-modulated radiotherapy), and the start and end dates of RT. The list of target volumes and organs at risk has been harmonized to have a homogeneous naming of each volume during extractions and analyses according to the following: CTVp_breast (Clinical Target Volume primary), CTVp_tumorbed, CTVp_thoracicwall, CTVn_interpectoralis (Clinical Target Volume nodal), CTVn_IMN (Internal Mammary Nodal), CTVn_L1, CTVn_L2, CTVn_L3, CTVn_L4, CTVn_Ltot, Heart, left anterior descending (LAD) coronary, Lung_right, Lung_left, Lungs, Humeral Head, Controlateral Breast, External, Spinal_cord, Thyroid, BrachialPlexus, and Esophagus.

Data were extracted, filtered and grouped according to sets of constraints by volumes (mean dose, median dose, DX% (dose covering X % of the volume expressed in Gy), VX Gy (volume receiving at least X Gy expressed in %), near-min dose, near-max dose).

Characteristics of the patients (age, medical history, clinical examination and concomitant treatments), tumors (including TNM, histology, HER2, estrogen and progesterone receptor), paraclinical examinations (blood/plasma tests, bone densitometry, cardiac echography or myocardial scintigraphy in case of treatment with anthracyclines/trastuzumab/RT to the left breast and/or Internal mammary chain), type of breast (lumpectomy, total mastectomy) and lymph node surgery (sentinel node, axillary dissection), chemotherapy, targeted anti-HER2 therapies and endocrine therapy were recorded from the CANTO data.

### 2.4. Data Management and Quality Control 

Quality control of clinical data was performed regarding RT data available (laterality, type of mammary and lymph node surgery) on Aquilab Onco Place™ versus December 2020 database lock of CANTO CRF (Case Report Form). All inconsistencies were corrected by the participating centers after reopening the files on the Aquilab Onco Place™ database before dose extractions. Quality control of dosimetric data was performed after a first extraction of Dmean and D95% of the volume CTVp_Breast or Chestwall for all patients with CTV delineated. We highlighted some dose inconsistencies and identified them by manually opening the dosimetry to understand their origin. A low dose away from the usually prescribed 50 Gy could indicate severe hypofractionation (used for partial breast irradiation protocols NCT01024582 and NCT01247233) or a dosimetry offset on the centering scanner (patient error or DICOM error).

### 2.5. Statistical Analysis 

We described characteristics and RT data available in CANTO-RT using parameters such as mean, median or inter quartile range (IQR) and the dispersion parameters as standard deviation (SD) and range for the quantitative variables, as well as the frequency (%) for the categorical variables (Appendix A: List of main variable) All analyses were conducted using SAS (Statistical Analysis System), version 9.4.

## 3. Results

### 3.1. CANTO-RT Characteristics

A total of 3875 BC patients matching all CANTO-RT inclusion and exclusion criteria, compliant with RT data check and with succeed analysis were selected among the 8708 patients treated with RT out of the 9599 CANTO patients. The CANTO-RT cohort included 1947 (50.2%) left-side, 1850 (47.8%) right-side and 78 (2%) bilateral BC patients with a median follow-up of 64 (range: 4 to 102) months. The baseline patient and tumor characteristics and treatment information are summarized in Table 1 and Table 2.

Many patients had cardiovascular risk factors: 650 (16.9%) active smoking, 190 (4.9%) type II diabetics, 904 (23.3%) hypertension, 500 (12.9%) dyslipidemia and 768 (19.8%) obesity. The vast majority of patients, 2586 (66.7%), had stage pT1; 2525 (65.2%) had pN0 disease; 3321 (85.7%) had hormone receptor-positive tumors; and 553 (14.3%) had human epidermal growth factor 2 (HER2)-positive tumors. Concerning systemic treatment, 2087 (53.8%) received neoadjuvant or adjuvant chemotherapy, 477 (12.3%) received adjuvant trastuzumab and 3138 (81%) received adjuvant endocrine therapy. Breast and lymph node surgery among the 3797 patients with unilateral RT were respectively breast conserving surgery in 3065 (80.4%) and total mastectomy in 747 (19.6%), sentinel node in 2712 (71.5%) and axillary dissection in 1080 (28.5%) patients. Concerning radiation therapy, tumor bed boost was delivered in 2658 patients (68.5%) and lymph node RT in 1356 patients (35%), including internal mammary chain in 844 patients (21.8%) and axillary level 1 (CTVn_L1) 284 (7.3%). Most patients, 3691 (95.3%), were treated with 3D conformal RT and 184 (4.7%) with IMRT. The vast majority of treatment, 2707 (69.9%), was normofractionated RT (50 Gy in 25 fractions in five weeks +/− boost on the tumor bed of 16 Gy in 8 fractions). Moderate hypofractionationated RT was delivered in 166 (4.3%) patients with mostly 40 Gy in 15 fractions of 2.67 Gy in three weeks. More severe hypofractionation (accelerated partial breast irradiation 38,5/40 Gy in 10 fractions or 30 Gy in 5 fractions) was used in 1.3%. The unspecified fractionation rate was 24.5% because of missing contours for CTV breast or chestwall (Table 3).

### 3.2. Summary of RT Data Available

An overview of the CANTO-RT comprehensive RT data in terms of target volumes and OAR available for dose extraction is provided in Table 3.

Regarding target volumes in patients where they were intended to be treated, 1999 (62.8%) CTV Breast were delineated after conservative surgery and 399 (52.3%) CTV Chestwall after total mastectomy. CTV tumor bed boost was delineated in 2457 (91.4%) patients. Regarding lymph node target volumes, 408 (29.9%) total lymph node volumes (CTVn_Ltot: summation of lymph node volumes), 53 (18.6%) axillary areas (CTVn_L1 and L2), 537 (48.9%) areas formerly called as subclaviculars (CTVn_L3) and 792 (58.4%) areas formerly called supraclavicular (CTVn_L4) were delineated. Internal mammary chains were delineated in 621 (73.2%) patients.

Regarding OAR volumes, heart was delineated in 2939 (75.8%) patients of whom anterior coronary artery was delineated in 188 (4.9%) patients. Among the 2939 heart contours available, 1100 (59.5%) were delineated in patients with right side BC, 1764 (90.1%) in patients with left side BC and 75 (96.2%) in patient with bilateral BC. Other volumes were not delineated in the same proportions in the patients respectively: 3492 (90.1%) right lung and 3470 (89.5%) left lung, 2197 (56.7%) spinal cord, 677 (17.5%) esophagus, 635 (16.4%) thyroid, contralateral breasts in about 7.5% patients with unilateral RT, humeral heads in 6% and brachial plexuses in 2.2%.

## 4. Discussion

CANTO-RT is one of the largest prospective multicenter cohorts of early breast cancer patients treated with RT including full DICOM RT and standardized longitudinal data. The CANTO-RT tumor characteristics were consistent with known contemporary epidemiology [7]. In our cohort, 3D conformal irradiation was the technique mostly used, whereas IMRT was limited during this period. The percentage of IMRT techniques is not homogeneous and varies by center, and the uptake of this technology stays unevenly spread around Europe [8]. Our series shows that depending on the treated side, OAR are not delineated in the same proportions. For example, heart was more often delineated to the left side (90.1%) than to the right side (59.5%), which probably shows a concern regarding the mean cardiac dose from irradiation of a left-sided breast cancer much higher than that for a right-sided breast cancer [9]. However, we know that depending on the anatomy, the dose to the heart, especially in cases of irradiation of the internal mammary nodal chain (IMN), is not null set even when treating right-sided BC [10]. We have also shown a heterogeneity of practice in the delineation rate of clinical target volumes (CTV) treated, which varied from 52% to 91% of the cases. The absence of delineation of a treated CTV didn’t allow for the proper appreciation of target volume coverage. As expected, tumor bed CTV had the highest rate of delineation (91%), while it was just the opposite for Chestwall CTV.

CANTO-RT has several strengths: It is one of the largest prospective multicenter cohorts in BC with full DICOM RT data ever published with the presence of a centralized database and is available on a single platform (Aquilab™) with innovative tools (Analytics). Second, CANTO-RT followed standard methodological quality criteria for observational studies [11,12]. The patient population has well-described inclusion and exclusion criteria: treatment information and patient-reported outcomes were reported with the use of standardized CRFs, and the length of observation has sufficient duration to apprehend treatment-related toxicity. Third, electronic transfer of DICOM data and quality control methods optimized the quality of RT data available, avoiding manual reporting of complex values to be found in a RT technical file. Thus, CANTO-RT reports on RT data available in one of the largest databases in the world with individual full DICOM RT files (CT, RT Structure, RT Dose, RT Plan) and with contemporary RT techniques. Initiatives to centralize information available on large-scale RT exist in some countries but not for a long duration, due to the technological challenges imposed by the volume of this data. The REQUITE cohort has recruited 4400 patients and is one of the largest multicenter cohorts of cancer patients treated with RT with standardized longitudinal data collection, but it mixes several tumor sites and is not specific to breast cancer (2057 patients) [13]. Other BC studies are retrospective (case-control study) and use outdated RT techniques with a reconstructed mean heart dose (MHD) derived from two-dimensional (2D) data using typical anatomy rather than individual CT-based information [14,15]. Unlike CANTO-RT, these studies are based on dosimetric estimates that are too imprecise to improve the assessment of the benefit/risk balance of RT in personalized medicine. In most trials, we just have the information of RT as yes/no. However, the evaluation of toxicity, volumes, doses, fractionation and techniques must be taken into account. Breast cancer treatments are multimodal and it is important to do analyses integrating the different treatment parameters to better understand the toxicities specific to each treatment and the links between them.

We admit some limitations. First, radiation therapy practices have already changed. Large, prospective and randomized phase III trials have demonstrated that hypofractionated treatment results in equivalent tumor controls, better or improved acute and late toxicity, better or improved breast cosmesis compared to conventionally-fractionated regimens for early-stage breast cancer [16,17,18]. Hypofractionated whole-breast irradiation has become the new standard of care for breast conservation therapy; preferred regimens are 40 Gy in 15 fractions. Caution should be taken when comparing trends in dose according to calendar years, since the change of fractionation regimens (from 50 Gy/25 to 40 Gy/15 and today 26 Gy/5 in some cases) will by itself lead to a reduction in physical dose. In addition, fractionation is unspecified for a significant rate of RT (24.5%) because of missing CTV breast or chest wall without the possibility of extracting the dose and deducing fractionation. Other practices were changing during the inclusion period, e.g., the tumor bed boost delivery, which is less prescribed in patients older than 50 or 60 [19], and IMRT techniques which are more often used nowadays as they have shown similar results in locoregional tumor control but show superior results in planning target volume coverage [20]. Then, the sub-group of patients selected for CANTO-RT was restricted to the top 10 recruiters for a convenience sample and could have introduced bias. Lastly, there are biases inherent in the delineation of OAR and target volumes during RT treatment planning: missing volumes and variability between the institutions and observers [21]. The guidelines for radiation therapy for early BC stay heterogeneous [22,23,24,25,26]. CANTO-RT could be a tool for comparing practices, and such international bases would be desirable in the future.

The use of this database will allow for the analysis of the dose–effect relationship of radiation received in the organs of women in the CANTO-RT cohort with a possible correlation to the toxicities graded during their prospective follow-up. There are several ongoing projects, such as heart, skin, lung toxicity analyses. CANTO-RT will try to improve knowledge on the relationship between RT toxicities and systemic treatments and the role of potential modifiers of this dose-response such as chemotherapy and hormonal therapy. Other objectives could be the use of statistics and artificial intelligence (Machine, Deep or/and Reinforcement Learning) combined with dosimetry reconstruction approaches to supplement the dosimetric data of the CANTO-RT database during collaborative projects. This cohort, with a large amount of data collected on characteristics, clinical, paraclinical, biological and RT data, will help improve the knowledge needed to develop personalized medicine for BC patients.

## 5. Conclusions

We successfully established CANTO-RT, a prospective cohort of 3875 early breast cancer patients with full individual clinical and DICOM RT data available showing an important heterogeneity in volumes contoured. CANTO-RT is a valuable resource, open for collaborative projects, for the identification and validation of clinical and dosimetric predictive factors of RT-related toxicities. Further long-term projects and follow up are ongoing, and we hope to expand the collection of RT data.

## Figures and Tables

**Figure 1 cancers-15-00751-f001:**
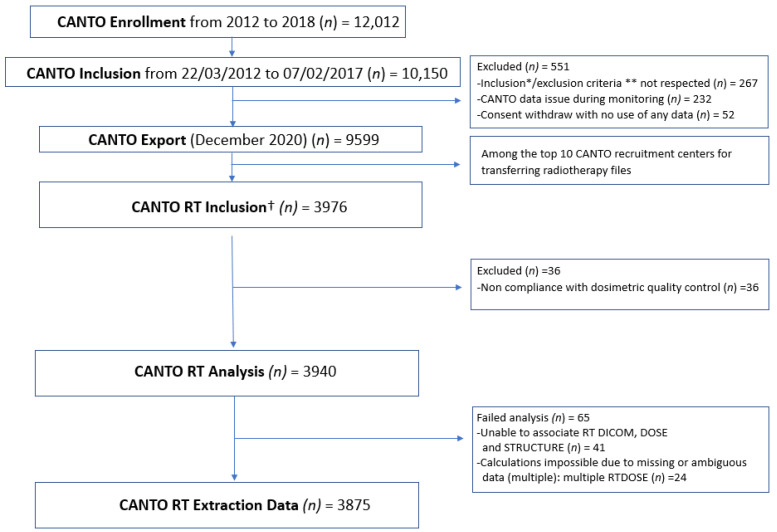
CANTO-RT Flowchart. RT: Radiation Therapy. * Inclusion criteria in CANTO: Female, 18 years of age and older, with infiltrating breast cancer diagnosed by cytology or histology, Tumor cT0-3, cN0-3, M0 before any treatment including surgery for breast cancer, patient fluent in French, free and informed consent for additional biological samples. † Inclusion criteria in CANTO RT: among the top 10 CANTO recruiting centers for transferring RT files to Aquilab, CT/RT in the same center + part of the selected centers, follow-up >3 years. ** Exclusion criteria in CANTO: Metastatic breast cancer; local recurrence of breast cancer; previous cancer within 5 years prior to cohort entry other than basal cell skin cancer or in situ cervical epithelioma; blood transfusion within the last 6 months; persons deprived of liberty or under guardianship (including curatorship).

**Figure 2 cancers-15-00751-f002:**
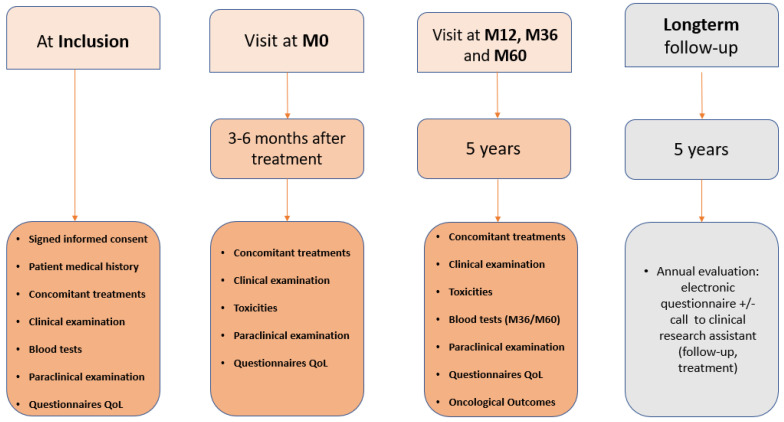
CANTO (Cancer Toxicities) cohort design.

**Figure 3 cancers-15-00751-f003:**
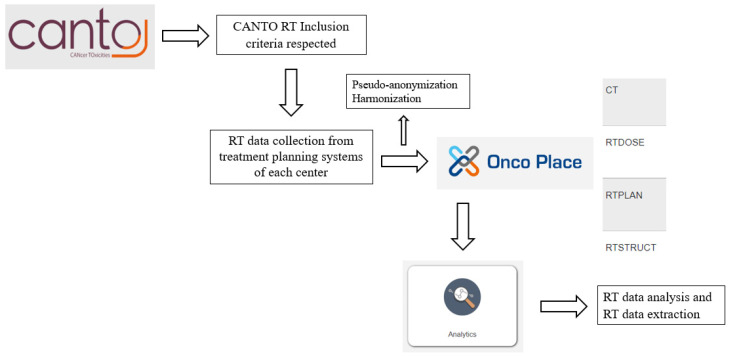
Radiotherapy (RT) Data Collection.

**Table 1 cancers-15-00751-t001:** Characteristics of the CANTO-RT patients.

Patient’s Characteristics	Breast Cancer Patients {N(%) or Mean (STD) or [Range]}
**Age at enrolment**	
*Mean (STD), [Range], years*	56.5 (11.2) [23.3–85.8]
**Smoking status at diagnosis**	
*Current*	650 (16.9)
*Former*	796 (20.5)
*Never*	2377 (61.3)
*Missing*	52 (1.3)
**Selected comorbidities**	
*Yes ** *Diabetes*	1566 (40.4) 190 (4.9)
*Hypertension*	904 (23.3)
*Dyslipidemia*	500 (12.9)
*BMI > 30* kg/m²	768 (19.8)
**Tumor size (pT)**	
*T0 ***	37 (1.0)
*T1*	2586 (66.7)
*T2*	1058 (27.3)
*T3*	177 (4.6)
*Missing*	17 (0.4)
**Nodal status (pN)**	
*0*	2525 (65.2)
*1*	1035 (26.7)
*2*	223 (5.8)
*3*	79 (2.0)
*Missing*	13 (0.3)
**Tumor histology**	
*Infiltrating Ductal*	3011 (77.7)
*Lobular*	473 (12.2)
*Others (including mixed)*	381 (9.8)
*Missing*	10 (0.3)
**Molecular subtype**	
*HR+ HER2+*	394 (10.2)
*HR+ HER2-*	2923 (75.4)
*HR- HER2+* *HR- HER2-* *Missing*	159 (4.1) 381 (9.8) 18 (0.5)
**SBR Grading**	
*I*	703 (18.1)
*II*	2019 (52.1)
*III*	1117 (28.8)
*Missing*	36 (0.9)
**Ki67**	
*No*	1657 (42.8)
*Yes*	1958 (50.5)
*<20%*	*1154 (58.9)*
*20–50%*	*657 (33.6)*
*>50%*	*147 (7.5)*
*Missing*	260 (6.7)

* At least one of selected comorbidity. ** Including ypT0. HR: Hormone Receptors. STD: Standard Deviation.

**Table 2 cancers-15-00751-t002:** Characteristics of the CANTO-RT treatments.

Treatment Characteristics	Breast Cancer Patients [N(%) or Mean (Range)]
**Type of chemotherapy**	
*No chemotherapy*	1788 (46.1)
*Neoadjuvant chemotherapy*	450 (11.6)
*Adjuvant chemotherapy*	1629 (42.0)
*Peri-adjuvant chemotherapy (neo + adjuvant)*	8 (0.2)
**Hormonal therapy**	
*No*	730 (18.8)
*Yes*	3138 (81)
*Missing*	7 (0.2)
**Trastuzumab treatment**	
*No or Not applicable*	3378 (87.2)
*Yes*	477 (12.3)
*Missing*	20 (0.5)
**Type of breast surgery**	
*Breast-conserving surgery*	3113 (80.3)
*Right*	*1488 (47.8)*
*Left*	*1577 (50.7)*
*Bilateral ***	*48 (1.5)*
*Total mastectomy*	734 (18.9)
*Right*	*359 (48.9)*
*Left*	*369 (50.3)*
*Bilateral ***	*6 (0.8)*
*Right breast-conserving surgery and left total mastectomy***	13 (0.3)
*Right total mastectomy and left breast-conserving ***	9 (0.2)
*None*	6 (0.2)
**Type of lymph node surgery**	
*Sentinel node*	2746 (70.9)
*Right sentinel node*	*1344 (48.9)*
*Left sentinel node*	*1368 (49.8)*
*Bilateral sentinel node ***	*34 (1.2)*
*Axillary dissection*	1086 (28.0)
*Right axillary dissection*	*506 (46.6)*
*Left axillary dissection*	*574 (52.9)*
*Bilateral axillary dissection ***	*6 (0.6)*
*Right sentinel node, Left axillary dissection ***	20 (0.5)
*Right axillary dissection, left sentinel node ***	12 (0.3)
*None*	11 (0.3)
**Radiation therapy**	
*Right Side*	1850 (47.8)
*Left Side*	1947 (50.2)
*Bilateral*	78 (2.0)
**Patients with boost**	
*No or Not applicable*	1217 (31.4)
*Yes*	2658 (68.6)
*Right Boost*	*1256 (47.3)*
*Left Boost*	*1344 (50.6)*
*Bilateral Boost ***	*31 (1.2)*
*Right Boost, no Left Boost ***	*16 (0.6)*
*Left Boost, no Right Boost ***	*11 (0.4)*
**Lymph node levels treated**	
*None*	2519 (65.0)
*Right*	*1222 (48.5)*
*Left*	*1258 (49.9)*
*Bilateral ***	*39 (1.5)*
*Yes*	1356 (35.0)
CTVn_L1	*284 (20.9)*
CTVn_L2	*340 (25.1)*
CTVn_L3	*1072 (79.1)*
CTVn_L4	*1348 (99.4)*
*Internal mammary chain*	*844 (62.2)*
*Right*	*404 (47.9)*
*Left*	*415 (49.2)*
*Bilateral ***	*4 (0.5)*
*Right only ***	*7 (0.8)*
*Left only ***	*14 (1.7)*
**Irradiation techniques**	
*3D*	3691 (95.3)
*IMRT*	184 (4.7)
** *Fractionation regimens* **	
*Normofractionation 25-fractions *¹*	2707 (69.9)
*Hypofractionation 15–16 fractions *²*	166 (4.3)
*Hypofractionation and Partial breast irradiation *³*	51 (1.3)
*Unspecified fractionation-CTV breast or chestwall not delineated ****	951 (24.5)

IMRT: Intensity modulated radiation therapy. ** Breast and lymph node surgery for patients with bilateral breast cancer and bilateral RT. *** No delineation of target volume allowing a dose extraction. **¹* 50 Gy/25 fractions/5 weeks +/- followed by a tumor boost of 16 Gy/8 fractions/1.5 week. **² 40,0 5Gy/15 fractions/3 weeks or 42.4 Gy/16 fractions/3.1 weeks* +/- followed by a tumor boost of 16 Gy/8 fractions/1.5 week. **³* Accelerated partial breast irradiation 10× 3,85 Gy or 10 × 4 Gy or 5 × 6 Gy.

**Table 3 cancers-15-00751-t003:** Radiotherapy data available in CANTO-RT.

	Number Delineated/Number Total	Volume Median (IQR), (cm^3^)	Dose Delivered, Mean (STD), (Gy)
**Target volumes**			
**CTV breast**	**62.8% (1999/3184)**	**598.0 (385.0–871.0)**	**53.6 (7.6)**
Right	62.0% (922/1488)	576.5 (371.0–845.0)	53.8 (7.0)
Left	63.2% (997/1577)	622.0 (398.0–905.0)	53.4 (8.3)
Bilateral–Right side	67.2% (41/61)	585.0 (384.0–866.0)	52.8 (4.8)
Bilateral–Left side	67.2% (39/58)	528.0 (387.0–766.0)	52.9 (4.2)
**CTV chestwall**	**52.3% (399/763)**	**314.0 (194.0–484.0)**	**48.9 (4.0)**
Right	51.0% (183/359)	314.0 (178.0–478.0)	49.1 (2.7)
Left	52.6% (194/369)	309.0 (204.0–489.0)	48.7 (5.0)
Bilateral–Right side	53.3% (8/15)	288.5 (205.0–445.5)	49.1 (3.4)
Bilateral–Left side	70.0% (14/20)	380.0 (243.0–521.0)	50.0 (1.2)
**CTV_tumorbed**	**91.4% (2457/2689)**	**20.8 (10.7–39.0)**	**64.8 (4.6)**
Right	91.5% (1149/1256)	19.6 (9.7–36.7)	64.8 (4.7)
Left	90.8% (1221/1344)	21.6 (11.6–40.5)	64.7 (4.6)
Bilateral–Right side	95.7% (45/47)	22.0 (15.9–39.7)	64 (2.8)
Bilateral–Left side	100.0% (42/42)	25.7 (10.4–45.2)	64 (3.1)
CTVn_Ltot ¹	29.9% (408/1364)	49.4 (33.3–78.2)	46.8 (7.5)
CTVn_L1 ²	18.6% (53/285)	53.8 (42.6–80.4)	46.9 (7.7)
CTVn_L2 ²	15.5% (53/342)	23.1 (16.9–48.6)	46.3 (7.7)
CTVn_L3 ²	48.9% (527/1077)	13.1 (7.8–19.8)	47.6 (5.4)
CTVn_L4 ²	58.4% (792/1356)	20.0 (13.5–28.1)	48.3 (4.8)
**CTV Internal mammary chain**	**73.2% (621/848)**	**4.6 (3–7.4)**	**46.5 (8.4)**
*Right*	73.0% (295/404)	4.9 (3.1–6.9)	46.3 (9.0)
*Left*	72.5% (301/415)	4.4 (2.9–7.7)	46.6 (7.7)
*Bilateral–Right side*	9.1% (1/11)	9.7 (9.7–9.7)	50.3 (.)
*Bilateral–Left side*	83.3% (15/18)	4.5 (3.6–6.6)	49.9 (1.6)
**Organs at risk**			
External Outline	98.3% (3810/3875)	20476.5 (17340.0–24588.0)	5.2 (2.1)
**Heart**	**75.8% (2939/3875)**	**609 (534.0–693.0)**	**3.4 (3.4)**
*Right*	59,5% (1100/1850)	**612.5 (538.0–697.5)**	**2.2 (2.8)**
*Left*	90,1% (1764/1947)	**604.0 (530.0–686.0)**	**4.0 (3.4)**
*Bilateral*	96,2% (75/78)	**651.6 (546.0–748.0)**	**6.0 (4.7)**
Lungs *	25.1% (972/3875)	2564.0 (2173.0–2979.5)	5.2 (3.5)
Right Lung	90.1% (3492/3875)	1428.5 (1226.0–1663)	5.1 (5.3)
Left Lung	89.5% (3470/3875)	1143.0 (961.0–1363.0)	5.3 (5.4)
Spinal Cord	56.7% (2197/3875)	49.0 (33.7–68.3)	1.7 (2.2)
Esophagus	17.5% (677/3875)	27.7 (22.4–33.9)	5.6 (6.4)
Thyroid	16.4% (635/3875)	13.0 (9.1–18.9)	12.9 (11.4)
LAD Coronary Artery	4.9% (188/3875)	5.3 (3.3–6.3)	15.0 (9.1)
Right Controlateral Breast	7.6% (147/1947)	7.6 (4.8–11.7)	2.7 (2.1)
Left Controlateral Breast	7.4% (137/1850)	7.8 (4.5–12.4)	2.9 (4.5)
Right Humeral Head	6.2% (119/1928)	45.3 (30.7–58.4)	10.3 (14.5)
Left Humeral Head	5.2% (105/2025)	47.4 (32.8–54.2)	6.7 (8.9)
Right Brachial Plexus	2.3% (45/1928)	8.9 (4.7–13.5)	28.6 (15.1)
Left Brachial Plexus	2.2% (44/2025)	7.6 (4.1–13.5)	24.4 (15.9)

IQR = First and third quartiles (75th and 25th percentiles); STD = Standard deviation; ^1^ Summation volume of all lymph nodes areas. ² According to the recommendations Estro 2015. * Right Lung and Left Lung were not always associated in Lung structure.

## Data Availability

Research data are stored in an institutional repository and will be shared upon request to the corresponding author.

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
