# Peer review of "CANTO-RT: One of the Largest Prospective Multicenter Cohort of Early Breast Cancer Patients Treated with Radiotherapy including Full DICOM RT Data"

_cancers, 2023, doi:10.3390/cancers15030751_

Round 1

Reviewer 1 Report

The manuscript entitled “CANTO-RT: One of the largest prospective multicentr cohort of early breast cancer patients treated with radiotherapy including full DICOM RT data” provides the methodology to collect data inducing full DICOM, quality control parameters, and patient characterization and the radiotherapy data in CANTO-RT. Overall, the manuscript is straightforward. The clear methodology and volume of data collect will provide valuable data regarding long-term toxicities in BCr patients treated with RT.

Line 112 – Please rephrase to include paraclinical parameters, assuming the blood chemistries, exams, etc are the same as described in ref 6. The sentence implies the organizational structure (Figure 1, ref 6), but it is unclear if it also implies the same paraclinical parameters. If this assumption is incorrect, please provide more details on the blood chemistries, toxicities, etc. being utilized.

Table 1 -  

·      Please include a Std for the age in addition to the mean(range)

·      For the comorbidities – is there overlap? Patients having more than 1 comorbidities.

·      For hormone receptors, ER, PR, HER2 – in ref 6 these patients are separated based on HR+/- and HER2 +/- - Is there a reason why this was not done here?

Line 234 – CTV delineated in …% of the cases – Assuming there is a percentage number that is missing.

Author Response

Responses to reviewers' comments

Reviewer #1

Comment #1 (R1C1): The manuscript entitled “CANTO-RT: One of the largest prospective multicentr cohort of early breast cancer patients treated with radiotherapy including full DICOM RT data” provides the methodology to collect data inducing full DICOM, quality control parameters, and patient characterization and the radiotherapy data in CANTO-RT. Overall, the manuscript is straightforward. The clear methodology and volume of data collect will provide valuable data regarding long-term toxicities in BCr patients treated with RT.

Response to comment 1: We thank the reviewer for these supportive comments.

Comment #2 (R1C2):  Line 112 – Please rephrase to include paraclinical parameters, assuming the blood chemistries, exams, etc are the same as described in ref 6. The sentence implies the organizational structure (Figure 1, ref 6), but it is unclear if it also implies the same paraclinical parameters. If this assumption is incorrect, please provide more details on the blood chemistries, toxicities, etc. being utilized.

Response to comment 2: You are perfectly right. Paraclinical parameters blood chemistries; exams, etc are the same as described in ref 6. Therefore, the following texts (Line 112) were reworded in the material and methods 2.2 data collection section:

“Patients and multimodal treatments characteristics as well as paraclinical parameters blood chemistries, exams, or toxicity data, etc were collected prospectively (Figure 2) and were the same as described in ref 6. Patients were  assessed at diagnosis (baseline), 3–6 (M0), 12 (M12), 36 (M36) and 60 (M60) months after completion of chemotherapy or RT whichever came last. In this study, we also collected the RT DICOM data (Digital Imaging and Communications in Medicine) whereas they were not used before. Computed tomography (CT) planning scans were performed in the treating center, DICOM RT data were pseudo-anonymized and stored on the UNITRAD online platform using AQUILAB Onco Place™.”

Comment #3 (R1C3): Table 1 -  

  • Please include a Std for the age in addition to the mean(range)

Response to comment 3: As suggested, we have included a Std for the age at enrolment in addition to the mean (range) in table 1.

Patients characteristics

Breast Cancer Patients {N(%) or Mean (STD) or [range]}

Age at enrolment

Mean (STD), years

56.5 (11.2)

Range, years

[23.3-85.8]

Comment #4 (R1C4): Table 1 -  

  • For the comorbidities – is there overlap? Patients having more than 1 comorbidities.

Response to comment 4: Yes, there is an overlap. In our series 40% of patients had at least one of selected cardiovascular risk factors at baseline. Table 1 was revised accordingly.

Selected comorbidities

Yes*

1566(40.4)

Diabetes

190 (4.9)

Hypertension

904 (23.3)

Dyslipidemia

500 (12.9)

BMI > 30kg/m²

768 (19.8)

              *At least one of selected comorbidity

Comment #5 (R1C5): Table 1 -  

  • For hormone receptors, ER, PR, HER2 – in ref 6 these patients are separated based on HR+/- and HER2 +/- - Is there a reason why this was not done here?

Response to comment 5: No, there is no particular reason; it was just another way to describe the data. In order to be consistent with previous publications, changes have been made in Table 1 as follows:

Molecular subtype+

HR+ HER2+

394 (10.2)

HR+ HER2-

2923 (75.4)

HR- HER2+

159 (4.1)

HR- HER2-

381 (9.8)

Missing

18 (0.5)

Comment #6 (R1C6): Line 234 – CTV delineated in …% of the cases – Assuming there is a percentage number that is missing.

Response to comment 6:

We thank the reviewer for his/her comment. The sentences were reworded as follows:

”We have also shown a heterogeneity of practice in the delineation rate of clinical target volumes (CTV) treated which varied from 52% to 91% of the cases. The absence of delineation of a treated CTV didn’t allow for proper appreciation of target volume coverage. As expected, tumor bed CTV had the highest rate of delineation (91%), while just the opposite for Chestwall CTV.”

Reviewer 2 Report

In this publication, the authors described a breast cancer database including the methodology to collect radiotherapy data and to ensure data quality control, and also they provided a first characterization of a study population from data extracted from this database.

General comment:

The topic is interesting because the analysis of big data is expected to be the mainstream of breast cancer research. Developing a breast cancer database can provide data and serve as valuable research tools. Depending on the research setting, the quality of the data is a major issue. Assuring that the data collection process does not contribute inaccuracies can help to assure the overall quality of subsequent analyses. In this study 26 french hospitals were involved which represents a large panel of patients and clinical practices.

When reading the publication, the description of the methodology to constitute the database and the rules for data quality control are not described in details.

Data management is a crucial point: it involves the implementation, administration of systems for the acquisition, storage and retrieval of data, protection by implementing high security levels. I would suggest to describe these steps. It would be also interesting to describe the data dictionary used primarily for data analysis including the list of variables in the database as well as the assigned variable names and a description of each type of variable (e.g., character, numeric, dates).Together with the database, it should provide comprehensive documentation that enables other researchers who might subsequently want to analyze the data to do so without any additional information.

Three tables dividing the information are shown: characteristics of patients, characteristics of treatments, and radiotherapy data. There are three types of relationships between these tables: one-to-one, one-to-many and many-to-many. It would be necesary to describe the process to bring the information together again in meaningful ways, based on clinical examples to illustrate how to find infor.

Data standardization is a key part of ensuring data quality. I would suggest to describe how the database was standardized (abbreviations, formatting during data entry, etc).

Additional comments:
Introduction:
Line 61-63: Could you please develop the sentence (or give some references) for readers? What are "DICOM data"? Could you please define "quality"

Materials and methods:
Line 109-110: Could you please describe what is the UNITRAD online platform and AQUILAB OncoPlace? What level of security to guarantee the privacy of data? Who owns the data?
Line 109: "pseudo-anonymized": what does it mean? why the data could not be completely anonymized? Has the Ethics Committee delivered a favourable opinion for this point?
Line 110: Could you please describe how did you assess data?
Line 118: Could you please explain what is the "Analytics dose module"?
Line 127: Could you please give the full text of the abbreviations at least once (CTVp, CTVn, IMN, L1, L2, L3, L4, etc)?
Line 134: What is the definition of "near-min dose" and "near-max dose"?
Line 146: Define what is "CRF"?
Line 159: Define what is "SAS"?

Results:
See my general comments. The results should show informations about the two first goals of the study: the methodology to collect radiotherapy data and to ensure data quality control.

Author Response

Responses to reviewers' comments

Reviewer #2

Comment #1 (R2C1): The topic is interesting because the analysis of big data is expected to be the mainstream of breast cancer research. Developing a breast cancer database can provide data and serve as valuable research tools. Depending on the research setting, the quality of the data is a major issue. Assuring that the data collection process does not contribute inaccuracies can help to assure the overall quality of subsequent analyses. In this study 26 french hospitals were involved which represents a large panel of patients and clinical practices.

When reading the publication, the description of the methodology to constitute the database and the rules for data quality control are not described in details.
Data management is a crucial point: it involves the implementation, administration of systems for the acquisition, storage and retrieval of data, protection by implementing high security levels. I would suggest to describe these steps.

Response to comment 1: We thank the reviewer for his interest as we share is opinion on big data such as provided by our work being the mainstream of breast cancer research.

All data necessary to the research were entered prospectively into the cohort case report forms (eCRFs). eCRF was completed by the investigator and other designated members duly acreditated from his/her staff. Data clarification forms (DCFs) were sent for data consistency validation, by the UNICANCER Central Data Center (CDC), under the responsibility of Mrs. Sophie Gourgou, located at the following address: Institut régional du Cancer Montpellier (ICM) / Val d’Aurelle, 208 rue des Apothicaires – Parc Euromédecine - 34298 Montpellier cedex 5 – France and Mrs Anna Laure Martin head of the data department of UNICANCER. Corrections addressed by these queries were made by persons authorized to complete the CRF. When using electronic CRF, traceability of access and changes were traced by the software (audit trial).

To ensure the authenticity and credibility of data in accordance with the “Décision portant sur les Bonnes Pratiques Cliniques, 24 November 2006”, the sponsor establishes a system of quality assurance that consists in:

  • The management and the monitoring of the cohort according to UNICANCER procedures;
  • The quality control data of the investigational centers by the monitor involves :
    • verifying that the cohort, as well as the current guidelines ICH-GCP, the national regulatory requirements, are accurately followed,
    • verifying the informed consent and the eligibility of each patient participating in another research,
    • verifying that the CRF data is consistent and in agreement with the source documents,
    • verifying the notification of each Serious Adverse Events (SAE),
    • verifying the drug (if any) traceability (dispatching, storage and accountability),
    • verifying (if applicable) that patients are not already participating in another research trial which may exclude their inclusion in this cohort. The monitor also verifies that patients have not participate in another trial following which an exclusion period if applicable before they can participate in another protocol,
  • The audit of the participating investigational centers when deemed necessary;

The monitors in charge of trial monitoring were mandated by UNICANCER. They had access to all patient data as necessary for their duty in accordance with the national regulatory requirements. The monitors were bound by professional secrecy under the national regulatory requirements. Written reports were issued to ensure monitoring visit traceability.

In order to ensure the optimal research quality control, the investigators committed to provide the monitor with direct access to all patient files.

As part of its audit program, UNICANCER audited investigational centers.
More generally, the investigator center and the investigator undertook to devote the time necessary to audit procedures, control and additional information requested by the sponsor or by a Concerned Competent Authority.

A Competent Authority can also conduct an inspection. If a Competent Authority requests an inspection, the investigator must inform the sponsor immediately that this request has been made. The investigator must provide a direct access to source documents.

The Cohort is conducted in accordance with the French national regulatory requirements, including but not limited to:

  • Loi Huriet (n° 88-1138) du 20 décembre 1988 relative à la Protection des Personnes se prêtant à la Recherche Biomédicale et modifiée par la loi d’orientation de Santé Publique (n° 2004-806) du 9 août 2004,
  • Loi Informatique et Libertés n° 78-17 du 6 janvier 1978 modifiée,
  • Loi de Bioéthique n° 2011-814 du 08 Juillet 2011,
  • Décision portant sur les Bonnes Pratiques Cliniques du 24 novembre 2006,

Prior to the start of the cohort, UNICANCER as sponsor has submitted the cohort protocol, patient information sheet(s), informed consent form(s), and other cohort-related documents as required by French national regulatory requirements, to the ANSM (Agence nationale de sécurité du médicament et des produits de santé) for authorization and to the CPP (Comité de protection des personnes)  for their written approval.

The sponsor has informed ANSM and the CPP, according to French national regulatory requirements, about protocol amendments including any substantial modification that require an ethical/regulatory reconsideration of the cohort protocol.

Data recorded in this cohort are subject to a computerized treatment by the UNICANCER Central Data Center in compliance with the “Loi Informatique et Libertés n° 78-17, 6 January 1978 modified” in agreement with European regulations GDPR.

The collection of biological samples implemented within the framework of the cohort was declared to ANSM in the same time that the request of Clinical Trial Authorization. Storage of the collection of biological samples is notified to the Minister of Research (and submitted to the CPP to notice if change of purpose of Research).

UNICANCER, the sponsor of the cohort certifies that it has taken out a Civil Liability insurance policy covering its civil liability for this clinical trial under its sponsorship. This insurance policy is in accordance with local laws and requirements. The insurance of the sponsor does not exempt the investigator and its team from maintaining their own liability insurance policy.

We propose to summarize this with the following sentence that we integrated in our manuscript in the material and methods section paragraph…:

“Canto was conducted in accordance with the French national regulatory requirements, good clinical practice guidelines and European General Data Protection Regulation (GDPR) as previously described by Vaz et al”( ref 6)

Comment #2 (R2C2):  It would be also interesting to describe the data dictionary used primarily for data analysis including the list of variables in the database as well as the assigned variable names and a description of each type of variable (e.g., character, numeric, dates). Together with the database, it should provide comprehensive documentation that enables other researchers who might subsequently want to analyze the data to do so without any additional information.
Response to comment 2:

The database collecting the study information for Canto is hosted by Unicancer. CANTO is open for collaborations so other researchers who might subsequently want to analyze the data can do so but only within a collaboration agreement as we need to previously inform all patients of the purpose of the use of their data to comply with regulation and protocol commitments. However, we have added a dictionary of variables used in the present study as a supplementary material in Supl Table 1.

Comment #3 (R2C3): Three tables dividing the information are shown: characteristics of patients, characteristics of treatments, and radiotherapy data. There are three types of relationships between these tables: one-to-one, one-to-many and many-to-many. It would be necesary to describe the process to bring the information together again in meaningful ways, based on clinical examples to illustrate how to find information.

Response to comment 3: We thank the reviewer for this comment. These 3 tables describe the whole CANTO-RT population and the treatments including radiotherapy that this population has received therefore no comparison between the tables is made. In Table 3 in particular, we have described the percentage of organs delineated and the doses delivered to targets and organs at risk for different clinical situations.

For information, we are currently working on studying the correlation of the average dose to the heart according to these different situations and cardiac toxicities. This research will be the subject of other publications.

Comment #4 (R2C4): Data standardization is a key part of ensuring data quality. I would suggest to describe how the database was standardized (abbreviations, formatting during data entry, etc).

Response to comment 4: Data were collected in a structured and standardized way in the Case Report Form, questionnaires were standard internationally validated questionnaires such has QLQC30. All radiotherapy data were collected in DICOM format. Digital Imaging and Communications in Medicine (DICOM) is the standard for the communication and management of medical imaging information and related data. DICOM is most commonly used for storing and transmitting medical images enabling the integration of medical imaging devices such as scanners, servers, workstations, printers, network hardware, and picture archiving and communication systems (PACS) from multiple manufacturers. It has been widely adopted .

In order to clarify this point the following sentence was added in our manuscript in section material and methods:

“Radiotherapy data were exported in standardized Digital Imaging and Communications in Medicine (DICOM) format by each investigating hospital to the UNITRAD platform hosted by Aquilab a company with health data hosting authorization. All data were automatically pseudo anonymized and converted to a homogeneous naming.”

Comment #5 (R2C5): Introduction:
Line 61-63: Could you please develop the sentence (or give some references) for readers? What are "DICOM data"? Could you please define "quality"

Response to comment 5: We thank the reviewer for this useful remark. Digital Imaging and Communications in Medicine (DICOM) for radiotherapy data is detailed in the abstract and in materials and methods Line 116 and involves « individual full DICOM RT data (CT, RT Structure, RT Dose, RT Plan) ». Quality refers to radiotherapy data (contoured volumes, dosimetry, etc.) for precise calculation and delivery of the planned dose.

The manuscript will be completed to give this information as follows: « However, toxicity factors related to RT and their consequences are poorly known because of limited DICOM data with limited analyses on contouring (target and organs at risk volumes), dose distribution, RT technique and quality involving precise calculation and delivery of the planned dose. »

Comment #6 (R2C6):  : Materials and methods:
Line 109-110: Could you please describe what is the UNITRAD online platform and AQUILAB OncoPlace? What level of security to guarantee the privacy of data? Who owns the data?

Response to comment 6: The UNITRAD online plateform (https://recherche.unicancer.fr/en/experts-groups/the-unicancer-group-of-translational-research-and-development-in-radiation-oncology-unitrad/) is a web-based secured platform to efficiently Share and Analyze Imaging and Radiotherapy data in our cohort. Using OncoPlace the Aquilab company software which automatically pseudonymize and structure DICOM radiotherapy data from each investigators site as it is compatible with all radiotherapy treatment planning systems. Aquilab and UNICANCER are accredited to host health data. As the sponsor of the cohort UNICANCER owns the data.

Comment #7 (R2C7): Line 109: "pseudo-anonymized": what does it mean? why the data could not be completely anonymized? Has the Ethics Committee delivered a favourable opinion for this point?

Response to comment 7: "Pseudo-anonymized" refers to nominative data for which the identity of the person has been replaced in the source file by a code, the correspondence between the code and the real identity being stored separately in a second file. This allows the lifting of anonymity or the study of correlations if necessary. Radiotherapy data cannot be fully anonymized as they include anatomical images that may contain specific anatomical characteristics of a specific patient (for example if a patient has a specific and rare prosthesis or implant). The Ethics committee approval was obtained as this is mandatory for such a cohort according to French regulation and was transmitted to the editor as previously requested (N° ID RCB: 2011-A01095-36). We send this in attachment to this letter for your information if needed.

Comment #8 (R2C8): Line 110: Could you please describe how did you assess data?

Response to comment 8: This issue has been partly dealt with as a response to reviewer 1 (R1C2). The obtaining of these data is detailed in reference 6 and explained in Figure 2. That’s why our sentence is modified as follow : « Patients and multimodal treatments characteristics as well as paraclinical parameters blood chemistries, exams, or toxicity data, etc were collected prospectively (Figure 2) and were the same as described in ref 6. Patients were assessed at diagnosis (baseline), 3–6 (M0), 12 (M12), 36 (M36) and 60 (M60) months after completion of chemotherapy or RT whichever came last. »

Comment #9 (R2C9): Line 118: Could you please explain what is the "Analytics dose module"?

Response to comment 9: We thank the reviewer for this useful remark. Analytics dose module is an asset for our methodology. This module allows you to extract histograms doses volumes and filter analyses according to many criteria. Analytics is a component of Onco Place (https://www.aquilab.com/products-services/clinical-trials/analytics/) that provides statistical analyses and data processing models based on data collected. Its purpose is to move beyond data collection, structuring and harmonization to reach data transformation and mining on cohorts of patients.

We’ll add this sentence for more details Line 132 :  « In Analytics Dose module, RT data were extracted, filtered and grouped according sets of constraints by volumes (mean dose, median dose, DX%: dose covering X % of the volume expressed in Gy, VX Gy : volume receiving at least X Gy expressed in % , near-min dose, near-max dose ) »

Comment #10 (R2C10): Line 127: Could you please give the full text of the abbreviations at least once (CTVp, CTVn, IMN, L1, L2, L3, L4, etc)?

Response to comment 10 : As suggested, we will give the abbreviations of CTVp (Clinical Target Volume Primary), CTVn (Clinical target volume nodal), and IMN (Internal Mammary Nodal)  at their first appearance. L1, L2, L3, L4 refers to axillary and supraclavicular node level. More details are noted Line 208-210. All volumes were named according to the European Society of Radiation Oncology (ESTRO) consensus ref 25 (Offersen Radiother Oncol. 2015).

Comment #11 (R2C11): Line 134: What is the definition of "near-min dose" and "near-max dose"?

Response to comment 11 : It refers to near-minimum absorbed dose i.e. the minimum dose of radiation received by 98% of the volume contoured (D98%) and near-maximum absorbed dose i.e the maximum radiation dose received by 2% of the volume contoured (D2%)

Comment #12 (R2C12):Line 146: Define what is "CRF"?

Response to comment 12 : It means « Case Report Form ». We will add the abbreviation.

Comment #13 (R2C13): Line 159: Define what is "SAS"?

Response to comment 13:  "Statistical Analysis System" (SAS)  is a statistical software suite developed by SAS Institute for data management, advanced analytics, multivariate analysis, business intelligence, criminal investigation and predictive analytics. https://www.sas.com/en_us/software/stat.html.

The software reference has been added to the manuscript.

Comment #14 (R2C14): Results:
See my general comments. The results should show informations about the two first goals of the study: the methodology to collect radiotherapy data and to ensure data quality control.

Response to comment 14: We thank the reviewer for this useful comment. Both methodology and quality control topics have been adressed within our response to the reviewers general comment in the first point of this letter. As suggested by the reviewer we have added our dictionary Table as Supl Table 1.

We also have changed the way we express our goal for a better understanding and presentation of the results :" In this paper, we describe the methodology used to collect RT data (full DICOM) and ensure RT data quality control to provide a first characterization of the study population and RT data in CANTO-RT."

Supl. Table 1 : dictionary

Variable name

Variable Label

assigned variable names

Variable Format

Age at enrolment

Age in year

Age_diag

numeric

Smoking status at diagnosis

1=Current; 2=Former ; 3=Never

TABAC_BI

character

Diabetes

0=No; 1=Yes;

Diabete_atcd

character

Hypertension

0=No; 1=Yes;

HTA_atcd

character

Dyslipidemia

0=No; 1=Yes;

dyslipidemie_atcd

character

BMI

kg/m2

IMC_BI

 numeric

Tumour size (pT)

0=T0; 1=T1; 2=T2; 3=T3

PTGlobal

character

Nodal status (pN)

0=0; 1=1; 3=2; 4=3

PN_M0

character

Tumour histology

1=Infiltrating Ductal; 2=Lobular; 3=Others (including mixed).

Tumour_histo

character

Hormone Receptors positive

0=Negative; 1=Positive;

HR

character

Estrogen receptors

0=Negative; 1=Positive;

RE

character

Progesterone receptors

0=Negative; 1=Positive;

RP

character

HER2

0=Negative; 1=Positive;

HER2

character

SBR Grading

1=I; 2=II; 3=III.

GRADE

character

Ki67

0=No; 1=Yes;

Ki67

character

Percent of Ki67

%

POURCKI67

numeric

Type of chemotherapy

0=No chemotherapy; 1=Neoadjuvant chemotherapy; 2=Adjuvant chemotherapy; 3=Peri-adjuvant chemotherapy (neo + adjuvant);

CHIMIOTYP

character

Hormonal therapy

0=No; 1=Yes;

Tumour_histo

character

Trastuzumab treatment

0=No; 1=Yes;

Base_herceptin

character

Lumpectomy

0=No; 1=Yes;

TUMOR_

character

Total mastectomy

0=No; 1=Yes;

MASTR_

character

Sentinel node

0=No; 1=Yes;

GANGLS_

character

Axillary dissection

0=No; 1=Yes;

CURAGE_

character

Radiation therapy

0=No; 1=Yes;

RADIO_

character

Patients with boost

0=No; 1=Yes;

Boost_

character

Lymph node levels treated

0=None; 1=Yes;

Aires_GG_

character

Level 1

0=None; 1=Yes;

I_all_

character

Level 2

0=None; 1=Yes;

II_all_

character

Level 3

0=None; 1=Yes;

III_all_

character

Level 4

0=None; 1=Yes;

IV_all_

character

Internal mammary chain

0=None; 1=Yes;

CMI_all_

character

Irradiation techniques

0=3D; 1=IMRT

Technique_

character

Fractionation regimens

0=Normofractionation 25-fractions; 1=Hypofractionation 15-16 fractions; 2=Hypofractionation and Partial breast irradiation;

Fractionnement__NF0__HF1_

character

Round 2

Reviewer 2 Report

Good work!